# Associations between sleep habits, performance in reading and mathematics, and inattention and hyperactivity,

**Katrina L. Grasby**[1,2,3], **Sally A. Larsen**[4]*, **Alice M. Gregory**[5], **Sarah Blunden**[6], **Katie S. Lewis**[7], **Juan J. Madrid-Valero**[8,9], **William L. Coventry**[10], **Brian Byrne**[10], **Richard K. Olson**[11]

**1** Queensland Institute of Medical Research, Herston, Queensland, Australia, **2** School of Biomedical Sciences, University of Queensland, St Lucia, Queensland, Australia, **3** School of Biomedical Sciences, Faculty of Health, Queensland University of Technology, Herston, Queensland, Australia, **4** School of Education, University of New England, Armidale, New South Wales, Australia, **5** Department of Psychology, Royal Holloway University of London, United Kingdom, **6** School of Health, Medical and Applied Sciences, Appleton Institute, Central Queensland University, South Australia, Australia, **7** School of Medicine, Cardiff University, Cardiff, United Kingdom, **8** Department of Human Anatomy and Psychobiology, University of Murcia, Spain, **9** Murcia Institute of Biomedical Research, IMIB-Arrixaca, Spain, **10** School of Psychology, University of New England, Armidale, New South Wales, Australia, **11** Psychology Department, University of Colorado, Boulder, Colorado, United States of America

* slarsen3@une.edu.au

## Abstract

This study examined relations between sleep, reading and mathematics, and inattention/hyperactivity in childhood and adolescence in a sample of Australian twins (maximum $n = 5524$; 51% female; 95% European ancestry; $M_{age}$ 8.6–14.6 years). Between and within models were used to assess if (a) differences between twins, or (b) differences between families, in five facets of sleep habits were associated with academic and behavioral measures. Some sleep measures uniquely predicted reading and mathematics in some grades, but overall, sleep quality, duration, bedtime regularity, daytime sleepiness, and snoring accounted for only ~1% of the academic variance. However, sleep measures accounted for up to 7% of the variance in inattention/hyperactivity. Post hoc analyses showed that inattention mediated most of the associations between sleep and reading and mathematics outcomes that were detected. This study adds to the literature showing small and inconsistent relations between sleep and academic performance in normative samples of children and adolescents.

## Introduction

This study explored associations between sleep, achievement in reading and mathematics, and inattentive and hyperactive behaviours in school-aged students. While a considerable body of research has examined relations between sleep and academic outcomes or sleep and Attention Deficit/Hyperactivity Disorder (ADHD) in childhood and adolescence, the current study expanded the focus to encompass all

**Data availability statement:** The data necessary to reproduce the analyses presented here are not publicly accessible due to restrictions placed by Australian governments on the use and re-use of the National Assessment Program: Literacy and Numeracy (NAPLAN) data. However, deidentified data and analytic code may be requested from the first and second authors, or from the University of New England (UNE) institutional Data Management office: Research at UNE (RUNE), contactable at rune@une.edu.au. As per UNE institutional policy, all research data will be stored at the RUNE repository at the conclusion of the active project from which the data were drawn (the Academic Development Study of Australian Twins). The survey materials necessary to replicate the findings are accessible from the second author or by contacting the RUNE office. The survey materials are publicly available and each survey form is recorded in the Supplementary file of the following paper: Larsen SA, Little CW, Grasby K, Byrne B, Olson RK, Coventry WL. The academic development study of Australian twins (ADSAT): Research aims and design. Twin Res Hum Genet. 2020;23(3):165–73. doi:10.1017/thg.2020.49 Survey forms are also cited in the current paper in the Methods section. Survey items can be found in the cited papers. Some of the test papers from the NAPLAN program are publicly accessible at this website: https://www.acara.edu.au/assessment/naplan/naplan-2012-2016-test-papers. Test papers in years subsequent to 2016 are not publicly available due to restrictions implemented by Australian governments on the national standardised assessment program test items. The analyses presented here were not preregistered.

**Funding:** Australian Research Council Discovery Project DP120102414 and DP150102441.

**Competing interests:** Alice Gregory is an advisor for a project initially sponsored by Johnson's Baby. She was a consultant for Perrigo (2021+). She receives royalties for two books Nodding Off (Bloomsbury Sigma, 2018) and The Sleepy Pebble (Flying Eye, 2019) and a sleep gift (The Gift of Sleep, Lawrence King Publishers, 2023). She was previously a CEO of Sleep Universal LTD (2022). She occasionally receives sample products related to sleep (e.g. blue light blocking glasses) and has given a paid talk to a business (Investec). She is a specialist subject editor at JCPP (sleep) for which she receives a small honorarium. She has contributed a paid article to Neurodiem.

three domains. In addition we examined these relations with subcomponents of sleep including bedtime regularity, sleep duration, sleep quality, daytime sleepiness, and sleep disordered breathing. Hereafter, unless specified, we use the terms *sleep habits* for behaviours at sleep time (e.g., timing, regularity, bedtime routines) and *sleep patterns* for specific subcomponents of sleep physiology (e.g., sleep duration and sleep quality or fragmentation).

Sleep patterns in school-aged children have been associated with variation in academic performance, with lower-quality sleep typically linked with poorer performance [1–4]. A meta-analysis by DeWald et al. [5] of up to 17 studies reported significant but modest overall effects, with around 1% of variation in school-based achievement explained by daytime sleepiness and sleep quality measures, and less by sleep duration. A more recent meta-analysis mainly reporting studies using student self-report measures of sleep and school performance [6] similarly documented significant but very modest effects of sleep quality on a variety of academic measures ($r = .089$), and a non-significant association with sleep duration. Additionally, Williamson et al. [4] showed that persistent sleep problems from infancy to middle childhood were associated with poorer performance in a composite measure of the same reading and mathematics assessments used in the present study. We considered it valuable to include reading and mathematics as separate outcomes in this study because other research has shown that sleep problems identified in childhood may impact some, but not all, neuropsychological functions in adolescence [7]. In some studies, an association between sleep habits and academic achievement has not been found, (e.g., [8]).

Likewise, sleep patterns have been shown to relate to a child's attentional capacity. For example, daytime behaviours of otherwise typically developing sleep-deprived children often mirror those of children with Attention Deficit / Hyperactivity Disorder (ADHD) diagnoses [9]. Three substantial reviews [10–12] reported that parents consistently rated children with diagnoses of ADHD as poorer sleepers compared with non-diagnosed children, though it is worth noting that this relationship is absent, or at least less evident, when more "objective" measures of sleep, such as actigraphs, are employed. A more recent review supported these findings [13], showing adolescents with ADHD report significantly more disturbed sleep patterns (*i.e.,* total sleep time) and experience more sleep problems compared with their typically developing peers. However, these patterns were evident only in studies using parent- or self-report; in the few studies that used more objective measures, the findings were not repeated.

Some of the evidence for the link between ADHD symptoms and sleep patterns is quasi-experimental, such as when adenotonsillectomy simultaneously reduces sleep-breathing difficulties and daytime inattention [14] (*c.f.* [11] for a summary). Some evidence is genuinely experimental, when intervention for sleep disorders in children with ADHD diagnoses improves both sleep quality and attention [15,16], or when experimentally shortened sleep leads to increased daytime sleepiness and inattention in students diagnosed with ADHD [17]. In other studies, however, relations between sleep quality and attention can be modest (or absent), and research often involves children who meet criteria for clinically disordered sleep and/or ADHD

[18]. Therefore, research is needed which does not select participants with clinically diagnosed ADHD or sleep problems in order to allow consideration of these associations across the full range of sleep habits and attention/hyperactivity behaviours.

Associations between ADHD and poorer academic performance are attested to in multiple studies, and a full review is not warranted in view of the consistency and pervasiveness of results in this domain [19,20]. Articles by Willcutt et al. [20,21] represent findings typical of, for example, the comorbidity of reading disability and ADHD [22], and the co-occurrence of reading and mathematics difficulties, along with a range of shared neuropsychological impairments. Prior results using the current sample confirm that reading and mathematics abilities correlate reasonably highly [23] and that both correlate negatively with inattention and hyperactivity (range $r = -.33 / -.49$) [24]. In view of a degree of inconsistency in the existing literature examining the relations between sleep habits and academic performance, and the lack of research that considers the role of inattention and/or hyperactivity in these associations, we considered it useful to present our results using a large, nation-wide sample of Australian school students.

The data we report here are drawn from a sample of children and adolescents across a six-year age range, and include parent-reported sleep habits and ratings of attention/ hyperactivity, and school-administered standardized reading and mathematics assessments taken nationwide by all Australian students. The sample comprises twins recruited as part of a longitudinal behavior-genetic study of academic achievement [25]. In our analyses we therefore use between and within models to account for the twin structure of the data and control for genetic and familial confounders [26] (further details in the Analysis Plan section). This study therefore aimed to contribute to a broader picture of the interrelatedness of sleep patterns, academic performance and behaviour by examining the associations between these domains in a large, community-based sample of child and adolescent twins.

## Materials and methods

Participants for this study were drawn from a large Australian-based longitudinal twin study [25]. Beginning in 2012 the study recruited a national sample of 2762 school-aged twin pairs (47% MZ; 51% Female). Participants were drawn from all Australian states and territories except the Northern Territory. The majority of the sample was of European ancestry (95%), and 82% of twins lived with both biological parents on entry to the study. Twins were eligible to participate in the study if they completed any of the Grades 3, 5, 7 or 9, in any calendar year beginning in 2008, and had therefore undertaken the National Assessment Program: Literacy and Numeracy (NAPLAN) tests in at least one grade. Children are typically aged 8–9 years at the time of the Grade 3 NAPLAN tests (Sample $M_{age} = 8.61$) and proceed through the tests at two-yearly intervals (see Table 1). Since the introduction of the NAPLAN in 2008, all Australian school students are expected to sit the standardized academic tests in each of the Grades 3, 5, 7 and 9. Thus twins who were recruited to the study in Grade 3, 2012, were followed up over the subsequent three biennial assessment grades. Those recruited when they were already in later grades (Grades 5, 7, or 9) provided retrospective, concurrent and follow-up data where applicable. All test results were sourced from state and territory education departments. Institutional ethical approval for the study was granted by the University of New England (HREC Approval #HE12–150 and #HE 18–163). Parents or guardians of twins provided written informed consent to participate in the study and for the researchers to request their childrens' NAPLAN scores from state or territory Education Departments. Recruitment of study participants began on September 1, 2012 and concluded on December 31, 2017.

The recruitment procedure meant that some variables were not collected for some grades. In particular, inattention and hyperactivity measures were only collected in the same year that twins sat their NAPLAN tests (*i.e.,* not retrospectively). Therefore, of the full subsamples at each grade, inattention and hyperactivity were reported for 742 twin pairs at Grade 3 (41%), 715 at Grade 5 (35%), 722 at Grade 7 (37%), and 657 at Grade 9 (38%). Parents reported on twins' sleep both concurrently (in the year of data collection) and retrospectively (in the year/s prior to data collection) where necessary, so missingness on these variables aligns with the questionnaire response rate (23–26% non-returned questionnaires).

 

**Table 1. Individual-level descriptive statistics by grade.**

| | N | Mean | SD | Females | MZ | DZSS | DZOS |
|---|---|---|---|---|---|---|---|
| **Grade 3** | | | | | | | |
| Biological Sex | 2518 | | | 1297 (51.5%) | | | |
| Zygosity | 2518 | | | | 1174 (46.6%) | 746 (29.6%) | 598 (23.7%) |
| SES[a] | 2518 | 0.11 | 1.05 | | | | |
| Age in years | 2518 | 8.61 | 0.39 | | | | |
| Reading[b] | 2489 | 457.54 | 85.75 | | | | |
| Mathematics[b] | 2485 | 429.19 | 74.22 | | | | |
| Inattention[c] | 1277 | 4.56 | 1.02 | | | | |
| Hyperactivity[c] | 1277 | 4.67 | 0.98 | | | | |
| Regular Bedtime[c] | 2518 | 6.06 | 1.51 | | | | |
| Duration School Nights[d] | 2518 | 6.65 | 0.84 | | | | |
| Duration Weekends[d] | 2518 | 6.33 | 0.91 | | | | |
| Sleep Quality[c] | 2518 | 6.05 | 1.14 | | | | |
| Daytime Sleepiness[c] | 2518 | 6.21 | 0.82 | | | | |
| Snoring[c] | 2518 | 5.90 | 1.55 | | | | |
| **Grade 5** | | | | | | | |
| Biological Sex | 2668 | | | 1377 (51.6%) | | | |
| Zygosity | 2668 | | | | 1238 (46.4%) | 792 (29.7%) | 638 (23.9%) |
| SES[a] | 2668 | 0.10 | 1.05 | | | | |
| Age in years | 2668 | 10.60 | 0.39 | | | | |
| Reading[b] | 2623 | 533.90 | 79.69 | | | | |
| Mathematics[b] | 2624 | 517.91 | 71.26 | | | | |
| Inattention[c] | 1239 | 4.63 | 1.04 | | | | |
| Hyperactivity[c] | 1239 | 4.81 | 1.02 | | | | |
| Regular Bedtime[c] | 2668 | 6.04 | 1.47 | | | | |
| Duration School Nights[d] | 2668 | 6.31 | 0.84 | | | | |
| Duration Weekends[d] | 2668 | 6.06 | 0.98 | | | | |
| Sleep Quality[c] | 2668 | 6.07 | 1.09 | | | | |
| Daytime Sleepiness[c] | 2668 | 6.19 | 0.80 | | | | |
| Snoring[c] | 2668 | 6.05 | 1.36 | | | | |
| **Grade 7** | | | | | | | |
| Biological Sex | 2538 | | | 1145 (51.3%) | | | |
| Zygosity | 2538 | | | | 1044 (46.7%) | 690 (31.0%) | 482 (22.4%) |
| SES[a] | 2538 | 0.03 | 1.08 | | | | |
| Age in years | 2538 | 12.58 | 0.42 | | | | |
| Reading[b] | 2481 | 580.98 | 67.71 | | | | |
| Mathematics[b] | 2475 | 579.19 | 71.38 | | | | |
| Inattention[c] | 1281 | 4.72 | 1.09 | | | | |
| Hyperactivity[c] | 1281 | 4.90 | 1.06 | | | | |
| Regular Bedtime[c] | 2538 | 5.90 | 1.52 | | | | |
| Duration School Nights[d] | 2538 | 5.76 | 0.92 | | | | |
| Duration Weekends[d] | 2538 | 5.78 | 1.04 | | | | |
| Sleep Quality[c] | 2538 | 6.06 | 1.14 | | | | |
| Daytime Sleepiness[c] | 2538 | 6.02 | 0.88 | | | | |
| Snoring[c] | 2538 | 6.04 | 1.41 | | | | |

*(Continued)*

**Table 1.** (Continued)

| | N | Mean | SD | Females | MZ | DZSS | DZOS |
|---|---|---|---|---|---|---|---|
| **Grade 9** | | | | | | | |
| Biological Sex | 2216 | | | 1145 (51.7%) | | | |
| Zygosity | 2216 | | | | 1044 (47.1%) | 690 (31.1%) | 482 (21.8%) |
| SES[a] | 2216 | 0.06 | 1.09 | | | | |
| Age in years | 2216 | 14.56 | 0.44 | | | | |
| Reading[b] | 2157 | 620.89 | 65.02 | | | | |
| Mathematics[b] | 2139 | 627.32 | 70.16 | | | | |
| Inattention[c] | 1163 | 4.86 | 1.10 | | | | |
| Hyperactivity[c] | 1161 | 5.05 | 1.08 | | | | |
| Regular Bedtime[c] | 2216 | 5.81 | 1.48 | | | | |
| Duration School Nights[d] | 2216 | 5.21 | 0.89 | | | | |
| Duration Weekends[d] | 2216 | 5.65 | 1.10 | | | | |
| Sleep Quality[c] | 2216 | 5.94 | 1.18 | | | | |
| Daytime Sleepiness[c] | 2216 | 5.66 | 1.05 | | | | |
| Snoring[c] | 2216 | 6.04 | 1.43 | | | | |

N = number of twin pairs; SES = socioeconomic status; MZ = monozygotic twins; DZSS = same sex dizygotic twins; DSOS = opposite sex dizygotic twins.

[a] Z-score from the full sample prior to selecting twins with relevant data.

[b] Scores range from 0–1000, all Grades are scored on the same scale.

[c]Scores range from 1–7, higher scores indicate more attention, less hyperactivity, more regular bedtime, better quality sleep, less daytime sleepiness, and less snoring.

[d]Scores range from 1–7 and represent number of hours sleep: 1 = less than 5; 2 = 5–6; 3 = 6–7; 4 = 7–8; 5 = 8–9; 6 = 9–10; 7 = 11 + hours.

Parents who did not return any follow up questionnaires for the study had lower average education and lower occupational prestige compared with parents who did return questionnaires, however there were no patterns relating to non-response after an initial questionnaire had been returned (*c.f.* [25] for additional detail on the sample characteristics). Proportion of retrospective reporting on sleep items for each grade is Grade 3 = 48%, Grade 5 = 36%, Grade 7 = 34%, Grade 9 = 28%. Since results in reading and mathematics subtests of the NAPLAN program were requested from state education departments, missingness occurred when students were unexpectedly absent from the testing day, withdrawn for any reason, or when education departments were unable to match student information with their records. For each grade we obtained assessment results for 95–97% of the sample.

### Measures

**Academic achievement.** The NAPLAN tests comprise five domains, reading comprehension, spelling, grammar, numeracy and writing. For this report we analyse only the reading comprehension (which we refer to as *reading*) and numeracy (*mathematics*) tests, since these two domains are fundamental to much of the school curriculum. Both the reading and mathematics tests are linked to the Australian National Curriculum. The reading test assessed students' ability to comprehend written English in a variety of text types (*e.g.,* narrative prose, media report, poetry), and interpret the use of language conventions in different written styles. The reading test comprised a series of multiple choice and written answer items, all scored either correct or incorrect. The mathematics tests assessed students' mathematical understanding, fluency, problem solving and reasoning. The tests assessed the three strands of the Australian Curriculum for mathematics: number and algebra, measurement and geometry, and statistics and probability. The mathematics tests comprised multiple-choice and text-entry answers, and the Grade 7 and Grade 9 tests contained a section allowing the

use of calculator. Published reliability statistics for the reading test range between α = 0.88–0.89 for each grade; and for the mathematics test, α = 0.86–0.93 [26,27]. Further information on test scope and item formats can be found on the National Assessment Program website (https://www.nap.edu.au/naplan).

For the duration of the study, NAPLAN testing was completed in May each year, approximately one third of the way through the Australian school year. Over the course of one week students sat papers for each of the assessed domains (the order of papers remained the same year-on-year). Scale scores for each test domain were generated using a Rasch modelling procedure [27]. A process of horizontal equating allowed scores to be compared as grades increase, and vertical equating ensured scores were mapped to the scale of the original equating test from 2008. The scaling and equating processes allowed us to collapse over multiple cohorts of students, even if they have completed alternative test forms in different calendar years. Scores in each NAPLAN domain range from 1–1000 and are interval scaled scores.

**Sleep habits and sleep patterns.** Parents reported the usual sleep habits and sleep patterns of each twin using 9 items. These were drawn from the 45 items of the Children's Sleep Habits Questionnaire (CHSQ) [28] to reflect the five distinct sleep dimensions detailed below. These domains aligned with those defined by the CHSQ (domain labels reported in brackets). First, for sleep habits, bedtime regularity was captured with the item *On school nights what time do the children go to bed, on average? How regular is this bedtime?* Parents responded on a seven point scale from *very irregular–very regular* (CHSQ: Bedtime Resistance). Second, for sleep patterns, four domains were measured. Sleep duration was assessed using two questions asking *How many hours of sleep do the children get on most school nights / weekend nights?* Parents estimated the number of hours on an eight-point scale ranging from *less than 5* to *11 + hours* (CHSQ: Sleep Duration). These items were treated separately in the analyses to allow for any marked differences in sleep duration on school nights compared with non-school nights [29]. Sleep quality relied on two items, *Do the children have difficulty getting to sleep?* and *Do the children wake up more than twice per night?* (CHSQ: combined Sleep Onset Delay and Night Wakings domains). Third, daytime sleepiness was measured by three items: *Do the children wake up in the morning and feel tired?*, *Do the children experience daytime sleepiness?,* and *Do the children fall asleep during daytime activities (for example, in class, in conversation, watching TV)?* (CHSQ: Daytime Sleepiness). Fourth, snoring was captured with the item *Do the children snore?* (CHSQ: Sleep Disordered Breathing). Parents responded to the two sleep quality, the three daytime sleepiness, and the snoring items on a seven-point scale ranging from *1 (never)* to *7 (always)*. For sleep quality (two items) and daytime sleepiness (three items), an average score was composed for each twin. Scores on sleep quality, daytime sleepiness, and snoring were reverse scored so that higher scores indicated better sleep quality, less snoring, and less daytime sleepiness.

The combination of these latter six items from the CHSQ were informed by the original arrangement of the CHSQ items into different domains, and by a preliminary exploratory factor analyses (EFA) using maximum likelihood estimation and oblique factor rotation. Sleep items at each grade were treated separately, and analyses were run for one randomly-selected twin and re-run with the second twin for comparison. We used the R statistical software [30] and the *pscyh* package [31] for EFAs. In all cases, a two-factor solution was considered most acceptable, taking into consideration scree plots, eigenvalues, item loadings and cross-loadings. In two-factor solutions, items consistently loaded only on one factor (or neither) with one item demonstrating cross-loadings above 0.30. The final factor solution explained 44–46% of the total variance, slightly differing by grade. This item, *Do the children wake up in the morning and feel tired?* was originally intended to capture Daytime Sleepiness in the CHSQ, so was retained in that factor. In the retained two-factor solution, the two items referencing CHSQ Sleep Onset Delay and Night Wakings loaded onto one factor, which we termed *sleep quality*; the three items from the CHSQ Daytime Sleepiness domain loaded onto one factor, termed *daytime sleepiness*, and the snoring item (CHSQ: Sleep Disordered Breathing) loaded on neither. The latter was therefore analysed as an individual item, rather than combined into a composite score. Code and results for the EFA are in Supporting Information S1 File.

**Inattention and hyperactivity.** Parents responded to the Strengths and Weaknesses of Attention-Deficit/Hyperactivity Disorder – Symptoms and Normal-Behavior rating scales (SWAN) [32] for each twin, each year the NAPLAN tests were

undertaken. The SWAN items are based on the then-current Diagnostic and Statistical Manual (4th ed.) diagnostic features of Attention-Deficit /Hyperactivity Disorder (ADHD). The SWAN is designed to provide information on childhood inattentive and hyperactive behaviour over a range from normal functioning to abnormal or problematic behaviour. Parents answered nine items for each twin for the inattention domain (e.g., *sustain attention on tasks or play activities*) and nine items for the hyperactivity domain (e.g., *await turn (stand in line and take turns)*). Responses were recorded on a seven-point scale ranging from *far below average* [1] to *average* [4] to *far above average [7]*. Items were then averaged within each domain to produce one inattention and one hyperactivity score, with lower ratings indicating greater inattention and higher hyperactivity. Prior research has demonstrated that the SWAN measure is invariant over the four measurement occasions, i.e., Grades 3, 5, 7 & 9; [33]. Coefficient alpha was calculated for each domain in each grade indicating good internal consistency reliability ($\alpha$ = .93−.95).

## Analysis plan

Between and within (BW) models were used to (a) assess if differences in sleep between twins *within* a pair were associated with the outcomes of interest, or (b) if differences in sleep *between* pairs/families were associated with the outcomes of interest. We use this design to control for familial confounders in samples of twin pairs. *i.e.,* genetic and shared environmental influences [34]. Complete twin pairs were selected from the full sample according to these criteria: they completed their NAPLAN tests in the same calendar year, were not missing data on age, sex, SES, zygosity, and the sleep variables, and had data on at least one of the outcomes of interest: reading, mathematics, inattention, or hyperactivity.

First, along with descriptive statistics, we report correlations between the sleep variables and each outcome variable (reading and mathematics, inattention and hyperactivity) at each grade. We used Griffin and Gonzalez's formula [35] to account for non-independence within twin pairs. Because of the sex effects suggested by DeWald et al. [5], we tested if sex should be included as a moderator in the models. We ran models with sex moderating all sleep predictors. For parsimony, where no significant sex moderation was found for an outcome, all sex moderation paths were dropped from the final model. Scatter plots with smoothed loess lines were inspected between the sleep variables and the outcomes to assess if non-linear terms should be included in the models. We checked for nonlinearity in these associations in case low quality of sleep (or even high levels, as in "too much" sleep) related differentially to academic performance. Prior to including opposite sex dizygotic twins, the correlations between the outcomes and the difference scores within twins on the sleep measures were measured separately for same-sex dizygotic males, same-sex dizygotic females, and opposite-sex dizygotic twins. These correlations were tested pairwise across all groups using Fisher's r to z transformation and a z-difference test.

Instead of assuming that the average difference in the outcome for a given difference in a predictor is the same when comparing two twins or two unrelated individuals, the between and within (BW) model estimates both of these effects in the same model [26]. The BW model estimates a coefficient for unrelated individuals, a between-pair beta ($\beta_B$), and a coefficient for differences between twins within a pair, a within-pair beta ($\beta_W$). The within-pair beta provides an estimate of the causal effect of change in predictor in the absence of non-shared confounders [36]. The general formula for this model is of the form:

$$Y_{ij} = \beta_0 + \beta_W\left(X_{ij} - \overline{X}_i\right) + \beta_B\overline{X}_i + \beta_C Covariate + \alpha_i + \varepsilon_{ij}$$

where $Y_{ij}$ is the outcome of interest and $X_{ij}$ is the exposure of interest for sibling $j$ within family $i$, $\overline{X}_i$ is the average of the exposure of interest for the twin pair in family $i$, $\beta_0$ is the intercept, $\beta_W$ is the within-pair coefficient, $\beta_B$ is the between-pair coefficient, $\beta_C$ is the fixed effect of a covariate, $\alpha_i$ is a random intercept for family nested within year, and $\varepsilon_{ij}$ is the individual random error term. This general formula was extended to include a within-pair and between-pair effect for each of the sleep exposures, essentially testing the effect for each sleep measure after controlling for the other sleep measures.

Demographic covariates were age, SES, sex, and cotwin sex. Age and SES were always shared by twins in a pair; however, this is not the case for sex. Sjölander et al. [37] described potential bias in $\beta_W$ due to collider effects if including a measured unshared covariate for one twin but not the cotwin; thus, cotwin sex was included as a covariate in models where sex was not shared by twins in a pair. For the reading and numeracy analyses, where a portion of the parental reports on sleep were retrospective, a covariate for this was included.

To test the significance of including all sleep variables in the model, a base model with covariates and random effects was run and compared to the full model with a likelihood-ratio test. The variance explained by all sleep variables was obtained by comparing the marginal $R^2$ of the full model with the base model. All analyses were run in $R$ (version 4.5.0). The package lme4 [38] was used to run the mixed models and we calculated cluster-robust standard errors using clubSandwich (CR2 estimator) [39] to account for the nested twin structure and potential influential data points.

Additionally, we applied a correction for multiple testing: Given four grades and four outcomes of interest, matrix spectral decomposition was implemented to estimate the effective number of independent variables [40]. A Bonferroni correction using the effective number of independent variables was made to adjust the alpha.

After running the planned analyses, a post-hoc BW mediation analysis was conducted to assess if the within-pair effects identified for reading and mathematics were mediated by within-pair inattention and/or hyperactivity. To assess this, the reading and mathematics BW models were re-run with a subsample that also had inattention and hyperactivity data to establish the total effect of each sleep-related predictor. Mediation was assessed using the product-of-coefficients method, where the indirect effect through each mediator was calculated as the product of: (a) the effect from predictor to the mediator, estimated via ordinary least squares regression, and (b) the effect of the mediator on the outcome controlling for the predictor and covariates, estimated via linear mixed-effects regression. Mediator models (for inattention and hyperactivity) included the same set of covariates, all sleep variables, and the between-pair inattention and hyperactivity variables. The outcome model included both mediators simultaneously to estimate their unique indirect effects. A random intercept for family was included in the outcome model. No random effect was included in the mediator models because the mediators were within-pair deviation scores. Indirect effects were estimated using a nonparametric cluster bootstrap (5,000 iterations), resampling families with replacement. At each iteration, the a and b paths were re-estimated and the product calculated. Confidence intervals (95%) were derived from the empirical bootstrap distribution.

## Results

### Descriptive statistics and correlations

The number of complete twin pairs with relevant data on at least one grade and outcome was 2127 (4254 individuals). Table 1 contains the descriptives for each grade. As noted above, data on inattention and hyperactivity were available on approximately half the sample in each grade because retrospective reports on these variables were not obtained. Histograms of the NAPLAN, inattention, and hyperactivity variables approximated a normal distribution, though most of the sleep variables were negatively skewed and leptokurtic. Additionally, the sleep, inattention, and hyperactivity variables were all measured on a Likert scale. Thus, Spearman's correlations were used to assess correlations among the sleep variables and the outcomes. Table 2 reports the correlations on the full sample, with those passing the multiple-testing threshold (*i.e., p* < .007) indicated with an asterisk. Sleeping longer on the weekends was positively associated with reading and maths in Grades 3, 5, and 7, and less snoring was associated with higher reading and maths scores in Grades 7 and 9. Unexpectedly, better quality sleep in Grade 7 was associated with lower reading scores. The significant correlations between the sleep variables and reading and maths outcomes were all small in size (varying in absolute value from .07 to .12).

Where significant, correlations between the sleep variables and attention and hyperactivity were slightly stronger than those for the reading and maths results but were still small in size (varying from .10 to .17). Better sleep quality was

**Table 2. Spearman correlations between sleep variables and reading, mathematics, attention, and hyperactivity.**

| Grade | Sleep Variables | Reading | | Mathematics | | Inattention[c] | | Hyperactivity[c] | |
|---|---|---|---|---|---|---|---|---|---|
| | | rs | p | rs | p | rs | p | rs | p |
| 3 | Regular Bedtime | −.01 | .573 | −.04 | .143 | .00 | .964 | .00 | .957 |
| | Duration on School Nights | .05 | .028 | .03 | .154 | .05 | .115 | .08 | .023 |
| | Duration on Weekends | .09* | <.001 | .08* | .001 | .09 | .008 | .11* | .002 |
| | Sleep Quality | −.02 | .368 | .03 | .200 | .13* | <.001 | .13* | <.001 |
| | Daytime Sleepiness | −.01 | .629 | .01 | .554 | .12* | <.001 | .09 | .009 |
| | Snoring | .04 | .080 | .03 | .117 | .03 | .335 | .00 | .892 |
| 5 | Regular Bedtime | −.05 | .046 | −.03 | .263 | .06 | .117 | .06 | .133 |
| | Duration on School Nights | .03 | .266 | .03 | .187 | .10* | .004 | .12* | .001 |
| | Duration on Weekends | .07* | .001 | .07* | .002 | .15* | <.001 | .17* | <.001 |
| | Sleep Quality | −.02 | .272 | .04 | .067 | .11* | <.001 | .15* | <.001 |
| | Daytime Sleepiness | .00 | .961 | .02 | .339 | .13* | <.001 | .09 | .008 |
| | Snoring | .06 | .009 | .05 | .021 | .04 | .253 | .02 | .483 |
| 7 | Regular Bedtime | −.02 | .411 | −.03 | .303 | .03 | .360 | .04 | .314 |
| | Duration on School Nights | −.03 | .247 | .02 | .351 | .02 | .636 | .00 | .951 |
| | Duration on Weekends | .08* | .001 | .08* | .002 | .04 | .192 | .01 | .728 |
| | Sleep Quality | −.07* | .001 | .03 | .213 | .10* | .001 | .13* | <.001 |
| | Daytime Sleepiness | −.03 | .282 | −.01 | .817 | .13* | <.001 | .14* | <.001 |
| | Snoring | .08* | <.001 | .08* | .001 | .12* | <.001 | .10* | .002 |
| 9 | Regular Bedtime | −.05 | .041 | −.05 | .040 | .07 | .040 | .08 | .029 |
| | Duration on School Nights | −.05 | .053 | .00 | .862 | .04 | .230 | .05 | .229 |
| | Duration on Weekends | .06 | .023 | .02 | .373 | .04 | .218 | .06 | .108 |
| | Sleep Quality | −.04 | .116 | .06 | .023 | .14* | <.001 | .17* | <.001 |
| | Daytime Sleepiness | −.04 | .115 | .02 | .436 | .17* | <.001 | .15* | <.001 |
| | Snoring | .12* | <.001 | .11* | <.001 | .14* | <.001 | .13* | <.001 |

*Significant correlation after correcting for multiple testing, p value <.007 [c] Scores range from 1–7, higher scores indicate more attention, less hyperactivity.

associated with less inattention and less hyperactivity in each grade. Similarly, less daytime sleepiness was associated with less inattention in each grade and less hyperactivity in Grades 7 and 9. Less snoring was also associated with less inattention and less hyperactivity in Grades 7 and 9. Longer sleep duration on both school nights and weekends was associated with less inattention and less hyperactivity in Grades 5 and longer sleep duration on weekends was associated with less hyperactivity in Grade 3.

**Examining assumptions.** Sex was a non-significant moderator of any sleep predictor for any outcome in any grade, therefore in the interests of parsimony, sex was not included as a moderator in the main analyses. Examination of plots revealed linear relations between the sleep variables and outcome variables. Thus, non-linear terms were not included in the models. Correlations between the two sleep duration variables, school nights and weekends, ranged from $r = .70$ in Grade 3 to $r = .40$ in Grade 9. Interestingly, this change in strength of correlation was linear, reducing by .1 for each biennial increase. Multicollinearity checks using variance inflation factor (VIF) were used to assess if it was acceptable to include both predictors. The highest VIF was 2.4 for sleep duration on a school night as a predictor of inattention in Grade 3 and the VIF tended to reduce with increasing grade. Females were coded 1, so the fixed effect for sex becomes the effect for females.

## Reading and mathematics outcomes

Taken together, the fixed effects for sleep explained a very small amount of variation in reading (ranging from 0.5–2.3%) and mathematics scores (ranging from 0.8–1.9%). Sleep duration on the weekend was a significant between-pair effects of reading in Grade 3, such that more sleep was associated with better reading. No other between-pair effect was significant for reading or mathematics in Grades 3 or 5. The within-pair effect for daytime sleepiness was significant, such that less tiredness during the day was associated with higher reading and mathematics scores in Grade 3. This within-pair effect controls for family-level confounds and the within-pair differences of the other sleep variables. Tables 3, 4 detail the fixed effects for the BW models for reading and mathematics, respectively. No between-pair or within-pair sleep effects were significantly related to either reading or mathematics in Grade 5. In Grade 7, the between-pair effects for sleep duration on the weekend and snoring were significant, such that more sleep on the weekend and less snoring were associated with higher reading scores, and less snoring was associated with higher mathematics scores. These between-pair effects are the average sleep duration or snoring score of twins in a pair, thus they reveal effects that are different between families. In Grade 9, the between-pair effect for less snoring continued to predict better reading and better mathematics scores. The within-pair effect for sleep were associated with Grade 9 reading and mathematics, such that better scores were predicted by better sleep quality. The within-pair effect for duration of sleep on a school night were associated with Grade 9 reading, such that better reading scores were predicted by less sleep on school nights.

## Inattention and hyperactivity

Taken together, the fixed effects for sleep explained a small amount of variation in inattention (ranging from 2.8–6.2%) and hyperactivity (ranging from 3.4–7.4%). The between-pair effect for sleep quality, such that better quality sleep (averaged for a twin pair) predicted better attention and less hyperactivity in every grade, except Grade 7 where it did not pass the multiple testing threshold. Tables 5, 6 detail the fixed effects for the BW models for inattention and hyperactivity, respectively. Regarding other between-pair sleep effects, in Grade 5 more sleep on the weekend predicted better attention and less hyperactivity, and in Grade 9 less daytime sleepiness predicted better attention. In Grade 3, the within-pair effects for daytime sleepiness were associated with inattention, such that individuals with less daytime sleepiness were predicted to have better attention. In Grade 9, better quality sleep predicted better attention and less hyperactivity. In Grade 5 and 7, no within-pair differences on sleep passed the multiple testing correction.

## Post-hoc analyses

In Grade 3, the within-pair effect for daytime sleepiness was associated with all four outcomes. Given that ADHD has been associated with poorer reading, a post-hoc analysis examined if the within-pair effect that linked daytime sleepiness with reading and mathematics might be mediated by within-pair differences in attention and/or hyperactivity. Mediation analyses confirmed that there was a significant indirect effect from within-pair differences in daytime sleepiness to reading and numeracy via within-pair differences in attention. The indirect effect of within-pair daytime sleepiness on reading via within-pair inattention was 7.08, 95% CI [3.24, 12.08]. No significant mediation occurred via within-pair differences in hyperactivity (indirect effect = −0.57, 95% CI [−3.47, 1.06]). The total effects and the direct effects for all predictors (*i.e.,* before and after including the inattention and hyperactivity variables in the models) are in Supporting Information S2 File. For Grade 3 reading, between-pair inattention was also a significant predictor of reading. Thus, less inattention (averaged for the pair) and less inattention for the individual were both unique predictors of better reading scores.

The results were similar for Grade 3 mathematics, the indirect effect of within-pair daytime sleepiness on mathematics via within-pair differences in inattention was 7.52, 95% CI [3.69, 12.27]. Again, within-pair hyperactivity was not a significant mediator (indirect effect = 0.08, 95% CI [−1.36, 1.47]). An unusual finding was that the between-pair effect for hyperactivity was also significant for mathematics; however this effect was in the opposite direction to expected, such that twin

**Table 3. Between and within models testing sleep measures as predictors of reading in grades 3, 5, 7, and 9.**

| Grade | Predictor | Std B | Std SE | t | df | p |
|---|---|---|---|---|---|---|
| **3** | Intercept | −0.01 | 0.04 | −0.22 | 522.3 | 0.827 |
| | SES | 0.32 | 0.02 | 14.16 | 503.0 | <0.001* |
| | Age | 0.12 | 0.02 | 5.08 | 408.7 | <0.001* |
| | Sex | 0.11 | 0.04 | 2.73 | 838.5 | 0.006 |
| | coTwin Sex | 0.03 | 0.04 | 0.79 | 837.9 | 0.431 |
| | Retrospective Sleep Report | −0.16 | 0.05 | −3.47 | 873.4 | 0.001 |
| | **Between-Pair Effects** | | | | | |
| | Regular Bedtime | 0.01 | 0.02 | 0.46 | 163.5 | 0.645 |
| | Duration on School Nights | −0.01 | 0.03 | −0.18 | 372.3 | 0.857 |
| | Duration on Weekends | 0.09 | 0.03 | 2.75 | 405.2 | 0.006 |
| | Sleep Quality | 0.01 | 0.03 | 0.54 | 233.0 | 0.589 |
| | Daytime Sleepiness | 0.02 | 0.03 | 0.57 | 204.4 | 0.569 |
| | Snoring | −0.01 | 0.03 | −0.52 | 304.6 | 0.603 |
| | **Within-Pair Effects** | | | | | |
| | Regular Bedtime | 0.03 | 0.02 | 1.32 | 11.2 | 0.214 |
| | Duration on School Nights | 0.01 | 0.02 | 0.53 | 58.3 | 0.599 |
| | Duration on Weekends | 0.00 | 0.02 | −0.24 | 35.2 | 0.813 |
| | Sleep Quality | −0.01 | 0.02 | −0.86 | 137.4 | 0.394 |
| | Daytime Sleepiness | 0.05 | 0.02 | 3.40 | 56.9 | 0.001 |
| | Snoring | 0.00 | 0.01 | −0.15 | 192.3 | 0.879 |
| **5** | Intercept | −0.06 | 0.04 | −1.50 | 560.9 | 0.134 |
| | SES | 0.33 | 0.02 | 14.97 | 543.1 | <0.001* |
| | Age | 0.08 | 0.02 | 3.29 | 449.9 | 0.001 |
| | Sex | 0.15 | 0.04 | 3.50 | 882.7 | <0.001* |
| | coTwin Sex | 0.01 | 0.04 | 0.24 | 883.6 | 0.808 |
| | Retrospective Sleep Report | −0.06 | 0.05 | −1.32 | 797.0 | 0.186 |
| | **Between-Pair Effects** | | | | | |
| | Regular Bedtime | −0.01 | 0.02 | −0.25 | 175.0 | 0.805 |
| | Duration on School Nights | 0.00 | 0.03 | 0.00 | 346.3 | 0.997 |
| | Duration on Weekends | 0.05 | 0.03 | 1.91 | 195.9 | 0.057 |
| | Sleep Quality | −0.01 | 0.03 | −0.28 | 184.4 | 0.779 |
| | Daytime Sleepiness | 0.00 | 0.02 | 0.15 | 234.4 | 0.883 |
| | Snoring | 0.04 | 0.03 | 1.69 | 265.3 | 0.092 |
| | **Within-Pair Effects** | | | | | |
| | Regular Bedtime | −0.01 | 0.02 | −0.33 | 19.1 | 0.745 |
| | Duration on School Nights | 0.00 | 0.02 | −0.17 | 93.3 | 0.866 |
| | Duration on Weekends | 0.00 | 0.02 | −0.01 | 58.6 | 0.990 |
| | Sleep Quality | 0.01 | 0.02 | 0.82 | 169.4 | 0.415 |
| | Daytime Sleepiness | 0.01 | 0.02 | 0.77 | 84.4 | 0.444 |
| | Snoring | 0.00 | 0.01 | 0.33 | 175.4 | 0.743 |

*(Continued)*

**Table 3.** (Continued)

| Grade | Predictor | Std B | Std SE | t | df | p |
|---|---|---|---|---|---|---|
| 7 | Intercept | −0.05 | 0.04 | −1.14 | 544.7 | 0.253 |
| | SES | 0.35 | 0.02 | 14.73 | 509.2 | <0.001* |
| | Age | 0.05 | 0.02 | 2.20 | 445.7 | 0.028 |
| | Sex | 0.07 | 0.04 | 1.70 | 815.1 | 0.090 |
| | coTwin Sex | −0.03 | 0.04 | −0.70 | 819.2 | 0.487 |
| | Retrospective Sleep Report | 0.08 | 0.05 | 1.51 | 754.2 | 0.131 |
| | **Between-Pair Effects** | | | | | |
| | Regular Bedtime | 0.05 | 0.02 | 2.11 | 185.9 | 0.036 |
| | Duration on School Nights | −0.07 | 0.03 | −2.27 | 394.8 | 0.024 |
| | Duration on Weekends | 0.09 | 0.03 | 3.06 | 370.0 | 0.002 |
| | Sleep Quality | −0.05 | 0.03 | −1.93 | 221.2 | 0.055 |
| | Daytime Sleepiness | −0.01 | 0.03 | −0.36 | 236.8 | 0.717 |
| | Snoring | 0.07 | 0.02 | 2.77 | 252.1 | 0.006 |
| | **Within-Pair Effects** | | | | | |
| | Regular Bedtime | 0.00 | 0.02 | 0.06 | 27.5 | 0.951 |
| | Duration on School Nights | −0.04 | 0.02 | −2.33 | 75.4 | 0.023 |
| | Duration on Weekends | 0.02 | 0.02 | 0.99 | 67.1 | 0.325 |
| | Sleep Quality | −0.01 | 0.02 | −0.51 | 159.1 | 0.611 |
| | Daytime Sleepiness | 0.02 | 0.02 | 1.03 | 98.0 | 0.306 |
| | Snoring | 0.00 | 0.01 | 0.12 | 177.5 | 0.907 |
| 9 | Intercept | −0.01 | 0.04 | −0.15 | 485.8 | 0.881 |
| | SES | 0.32 | 0.03 | 12.28 | 451.2 | <0.001* |
| | Age | 0.03 | 0.02 | 1.12 | 395.5 | 0.263 |
| | Sex | 0.06 | 0.05 | 1.29 | 657.9 | 0.197 |
| | coTwin Sex | −0.11 | 0.05 | −2.39 | 658.1 | 0.017 |
| | Retrospective Sleep Report | 0.12 | 0.05 | 2.12 | 552.6 | 0.035 |
| | **Between-Pair Effects** | | | | | |
| | Regular Bedtime | 0.01 | 0.03 | 0.47 | 177.7 | 0.636 |
| | Duration on School Nights | −0.05 | 0.03 | −1.78 | 313.1 | 0.076 |
| | Duration on Weekends | 0.04 | 0.03 | 1.45 | 344.1 | 0.148 |
| | Sleep Quality | −0.02 | 0.03 | −0.70 | 256.4 | 0.483 |
| | Daytime Sleepiness | −0.06 | 0.03 | −2.00 | 270.8 | 0.047 |
| | Snoring | 0.12 | 0.03 | 4.16 | 211.2 | <0.001* |
| | **Within-Pair Effects** | | | | | |
| | Regular Bedtime | 0.01 | 0.01 | 1.00 | 45.1 | 0.322 |
| | Duration on School Nights | −0.05 | 0.02 | −2.85 | 100.2 | 0.005 |
| | Duration on Weekends | −0.01 | 0.01 | −0.62 | 77.6 | 0.539 |
| | Sleep Quality | 0.04 | 0.02 | 2.73 | 173.4 | 0.007 |
| | Daytime Sleepiness | 0.01 | 0.02 | 0.34 | 110.2 | 0.736 |
| | Snoring | 0.02 | 0.02 | 1.15 | 152.5 | 0.251 |

*Indicates significant predictor after correcting for multiple testing, p value < .007.

**Table 4. Between and within models testing sleep measures as predictors of mathematics in grades 3, 5, 7, and 9.**

| Grade | Predictor | Std B | Std SE | t | df | p |
|---|---|---|---|---|---|---|
| 3 | Intercept | 0.12 | 0.04 | 2.82 | 522.3 | 0.005 |
| | SES | 0.30 | 0.02 | 12.75 | 502.8 | <0.001* |
| | Age | 0.12 | 0.02 | 4.66 | 403.7 | <0.001* |
| | Sex | −0.21 | 0.04 | −4.89 | 896.3 | <0.001* |
| | coTwin Sex | 0.01 | 0.04 | 0.13 | 896.0 | 0.896 |
| | Retrospective Sleep Report | −0.03 | 0.05 | −0.70 | 872.6 | 0.481 |
| | **Between-Pair Effects** | | | | | |
| | Regular Bedtime | −0.01 | 0.02 | −0.46 | 163.6 | 0.647 |
| | Duration on School Nights | −0.01 | 0.03 | −0.34 | 371.8 | 0.735 |
| | Duration on Weekends | 0.09 | 0.03 | 2.54 | 404.7 | 0.012 |
| | Sleep Quality | 0.02 | 0.03 | 0.83 | 232.7 | 0.408 |
| | Daytime Sleepiness | 0.03 | 0.03 | 1.13 | 204.1 | 0.258 |
| | Snoring | −0.01 | 0.03 | −0.43 | 304.4 | 0.667 |
| | **Within-Pair Effects** | | | | | |
| | Regular Bedtime | 0.00 | 0.02 | −0.16 | 11.1 | 0.877 |
| | Duration on School Nights | 0.02 | 0.02 | 0.87 | 58.0 | 0.390 |
| | Duration on Weekends | −0.02 | 0.02 | −1.10 | 35.4 | 0.280 |
| | Sleep Quality | 0.00 | 0.02 | 0.14 | 137.7 | 0.885 |
| | Daytime Sleepiness | 0.04 | 0.01 | 3.05 | 57.3 | 0.003 |
| | Snoring | −0.01 | 0.01 | −0.73 | 191.8 | 0.468 |
| 5 | Intercept | 0.09 | 0.04 | 2.15 | 562.5 | 0.032 |
| | SES | 0.33 | 0.02 | 14.29 | 542.2 | <0.001* |
| | Age | 0.08 | 0.02 | 3.43 | 449.3 | 0.001 |
| | Sex | −0.25 | 0.04 | −6.35 | 978.8 | <0.001* |
| | coTwin Sex | 0.05 | 0.04 | 1.31 | 976.7 | 0.190 |
| | Retrospective Sleep Report | 0.04 | 0.05 | 0.89 | 797.9 | 0.372 |
| | **Between-Pair Effects** | | | | | |
| | Regular Bedtime | −0.01 | 0.02 | −0.42 | 174.6 | 0.677 |
| | Duration on School Nights | −0.01 | 0.03 | −0.23 | 346.2 | 0.822 |
| | Duration on Weekends | 0.05 | 0.03 | 1.87 | 195.8 | 0.063 |
| | Sleep Quality | 0.03 | 0.03 | 1.26 | 184.2 | 0.209 |
| | Daytime Sleepiness | 0.00 | 0.02 | 0.14 | 234.5 | 0.887 |
| | Snoring | 0.05 | 0.03 | 1.80 | 265.4 | 0.073 |
| | **Within-Pair Effects** | | | | | |
| | Regular Bedtime | 0.01 | 0.01 | 0.53 | 19.1 | 0.603 |
| | Duration on School Nights | −0.03 | 0.02 | −2.00 | 94.7 | 0.048 |
| | Duration on Weekends | 0.03 | 0.01 | 1.90 | 59.0 | 0.063 |
| | Sleep Quality | 0.01 | 0.01 | 0.84 | 168.7 | 0.403 |
| | Daytime Sleepiness | 0.03 | 0.02 | 1.57 | 84.4 | 0.121 |
| | Snoring | −0.01 | 0.01 | −0.99 | 175.5 | 0.322 |

*(Continued)*

**Table 4.** (Continued)

| Grade | Predictor | Std B | Std SE | t | df | p |
|---|---|---|---|---|---|---|
| 7 | Intercept | 0.12 | 0.04 | 2.99 | 541.7 | 0.003 |
| | SES | 0.37 | 0.02 | 14.85 | 506.7 | <0.001* |
| | Age | 0.02 | 0.02 | 0.81 | 443.4 | 0.419 |
| | Sex | −0.26 | 0.04 | −6.30 | 853.7 | <0.001* |
| | coTwin Sex | −0.03 | 0.04 | −0.77 | 856.3 | 0.443 |
| | Retrospective Sleep Report | 0.09 | 0.05 | 1.76 | 754.4 | 0.079 |
| | **Between-Pair Effects** | | | | | |
| | Regular Bedtime | 0.02 | 0.02 | 0.77 | 185.5 | 0.445 |
| | Duration on School Nights | −0.03 | 0.03 | −1.16 | 393.9 | 0.248 |
| | Duration on Weekends | 0.06 | 0.03 | 2.16 | 368.7 | 0.032 |
| | Sleep Quality | 0.04 | 0.03 | 1.57 | 220.1 | 0.118 |
| | Daytime Sleepiness | −0.05 | 0.03 | −1.75 | 236.5 | 0.082 |
| | Snoring | 0.06 | 0.02 | 2.75 | 254.6 | 0.006 |
| | **Within-Pair Effects** | | | | | |
| | Regular Bedtime | 0.00 | 0.02 | 0.01 | 26.7 | 0.991 |
| | Duration on School Nights | 0.00 | 0.02 | 0.11 | 77.6 | 0.912 |
| | Duration on Weekends | 0.00 | 0.02 | −0.17 | 67.5 | 0.864 |
| | Sleep Quality | 0.00 | 0.01 | 0.17 | 156.9 | 0.863 |
| | Daytime Sleepiness | 0.02 | 0.01 | 1.64 | 98.6 | 0.105 |
| | Snoring | 0.00 | 0.01 | −0.12 | 177.5 | 0.902 |
| 9 | Intercept | 0.15 | 0.04 | 3.23 | 482.8 | 0.001 |
| | SES | 0.35 | 0.03 | 13.35 | 449.2 | <0.001* |
| | Age | 0.00 | 0.03 | 0.08 | 394.6 | 0.933 |
| | Sex | −0.30 | 0.05 | −6.24 | 692.3 | <0.001* |
| | coTwin Sex | 0.00 | 0.05 | 0.03 | 689.9 | 0.976 |
| | Retrospective Sleep Report | 0.03 | 0.05 | 0.50 | 549.3 | 0.615 |
| | **Between-Pair Effects** | | | | | |
| | Regular Bedtime | −0.01 | 0.03 | −0.42 | 174.2 | 0.672 |
| | Duration on School Nights | −0.01 | 0.03 | −0.43 | 309.0 | 0.666 |
| | Duration on Weekends | 0.00 | 0.03 | −0.08 | 343.5 | 0.936 |
| | Sleep Quality | 0.04 | 0.03 | 1.44 | 255.3 | 0.151 |
| | Daytime Sleepiness | −0.03 | 0.03 | −0.97 | 270.3 | 0.333 |
| | Snoring | 0.10 | 0.03 | 3.76 | 209.2 | <0.001* |
| | **Within-Pair Effects** | | | | | |
| | Regular Bedtime | 0.02 | 0.02 | 1.31 | 44.8 | 0.197 |
| | Duration on School Nights | −0.04 | 0.02 | −2.42 | 98.5 | 0.018 |
| | Duration on Weekends | −0.03 | 0.02 | −1.97 | 78.3 | 0.052 |
| | Sleep Quality | 0.04 | 0.02 | 2.72 | 171.5 | 0.007 |
| | Daytime Sleepiness | 0.02 | 0.02 | 1.43 | 110.9 | 0.156 |
| | Snoring | 0.02 | 0.01 | 1.40 | 148.9 | 0.164 |

* Indicates significant predictor after correcting for multiple testing, p value < .007

**Table 5. Between and within models testing sleep measures as predictors of inattention in grades 3, 5, 7, and 9.**

| Grade | Predictor | Std B | Std SE | t | df | p |
|---|---|---|---|---|---|---|
| **3** | Intercept | −0.19 | 0.05 | −3.63 | 300.9 | <0.001* |
| | SES | 0.16 | 0.03 | 4.77 | 261.5 | <0.001* |
| | Age | 0.11 | 0.04 | 3.01 | 221.9 | 0.003* |
| | Sex | 0.43 | 0.06 | 7.06 | 437.9 | <0.001* |
| | coTwin Sex | −0.06 | 0.06 | −0.96 | 437.5 | 0.336 |
| | **Between-Pair Effects** | | | | | |
| | Regular Bedtime | 0.00 | 0.04 | 0.12 | 81.4 | 0.905 |
| | Duration on School Nights | −0.01 | 0.05 | −0.20 | 194.9 | 0.843 |
| | Duration on Weekends | 0.00 | 0.05 | 0.05 | 215.3 | 0.964 |
| | Sleep Quality | 0.13 | 0.04 | 3.15 | 148.6 | 0.002* |
| | Daytime Sleepiness | 0.08 | 0.04 | 1.81 | 127.2 | 0.073 |
| | Snoring | −0.02 | 0.04 | −0.52 | 174.2 | 0.606 |
| | **Within-Pair Effects** | | | | | |
| | Regular Bedtime | 0.05 | 0.04 | 1.26 | 4.9 | 0.265 |
| | Duration on School Nights | 0.03 | 0.03 | 0.97 | 28.7 | 0.343 |
| | Duration on Weekends | −0.01 | 0.03 | −0.38 | 18.5 | 0.709 |
| | Sleep Quality | 0.06 | 0.02 | 2.40 | 81.6 | 0.019 |
| | Daytime Sleepiness | 0.10 | 0.02 | 4.17 | 56.7 | <0.001* |
| | Snoring | −0.06 | 0.02 | −2.61 | 116.3 | 0.010 |
| **5** | Intercept | −0.24 | 0.05 | −4.60 | 284.9 | <0.001* |
| | SES | 0.12 | 0.04 | 3.30 | 249.2 | 0.001* |
| | Age | 0.05 | 0.04 | 1.25 | 238.5 | 0.212 |
| | Sex | 0.54 | 0.06 | 8.69 | 446.9 | <0.001* |
| | coTwin Sex | −0.07 | 0.06 | −1.20 | 446.7 | 0.232 |
| | **Between-Pair Effects** | | | | | |
| | Regular Bedtime | 0.06 | 0.03 | 1.65 | 81.4 | 0.102 |
| | Duration on School Nights | −0.03 | 0.04 | −0.79 | 178.4 | 0.430 |
| | Duration on Weekends | 0.13 | 0.04 | 3.48 | 67.8 | 0.001* |
| | Sleep Quality | 0.11 | 0.04 | 2.58 | 107.1 | 0.011 |
| | Daytime Sleepiness | 0.08 | 0.04 | 1.95 | 126.8 | 0.053 |
| | Snoring | 0.02 | 0.03 | 0.55 | 133.2 | 0.584 |
| | **Within-Pair Effects** | | | | | |
| | Regular Bedtime | 0.05 | 0.03 | 1.60 | 9.5 | 0.143 |
| | Duration on School Nights | −0.02 | 0.03 | −0.72 | 74.1 | 0.471 |
| | Duration on Weekends | 0.02 | 0.02 | 0.76 | 34.5 | 0.452 |
| | Sleep Quality | 0.04 | 0.02 | 1.83 | 99.8 | 0.070 |
| | Daytime Sleepiness | 0.05 | 0.03 | 2.07 | 48.8 | 0.043 |
| | Snoring | −0.02 | 0.02 | −1.02 | 90.5 | 0.312 |

*(Continued)*

**Table 5.** (Continued)

| Grade | Predictor | Std B | Std SE | t | df | p |
|---|---|---|---|---|---|---|
| 7 | Intercept | −0.14 | 0.05 | −2.68 | 301.3 | 0.008 |
| | SES | 0.17 | 0.03 | 4.95 | 266.3 | <0.001* |
| | Age | 0.01 | 0.04 | 0.37 | 232.3 | 0.711 |
| | Sex | 0.46 | 0.06 | 7.37 | 477.4 | <0.001* |
| | coTwin Sex | −0.19 | 0.06 | −2.97 | 477.2 | 0.003* |
| | **Between-Pair Effects** | | | | | |
| | Regular Bedtime | 0.02 | 0.03 | 0.63 | 99.5 | 0.533 |
| | Duration on School Nights | −0.04 | 0.04 | −0.91 | 177.6 | 0.366 |
| | Duration on Weekends | 0.03 | 0.04 | 0.74 | 181.3 | 0.461 |
| | Sleep Quality | 0.09 | 0.04 | 2.25 | 181.8 | 0.026 |
| | Daytime Sleepiness | 0.04 | 0.04 | 1.14 | 131.0 | 0.256 |
| | Snoring | 0.05 | 0.04 | 1.37 | 140.1 | 0.173 |
| | **Within-Pair Effects** | | | | | |
| | Regular Bedtime | 0.03 | 0.02 | 1.50 | 15.2 | 0.154 |
| | Duration on School Nights | −0.03 | 0.03 | −1.33 | 47.8 | 0.190 |
| | Duration on Weekends | 0.03 | 0.03 | 1.17 | 56.9 | 0.248 |
| | Sleep Quality | 0.02 | 0.03 | 0.83 | 92.8 | 0.407 |
| | Daytime Sleepiness | 0.06 | 0.03 | 2.24 | 70.9 | 0.028 |
| | Snoring | 0.00 | 0.02 | −0.13 | 96.2 | 0.899 |
| 9 | Intercept | −0.24 | 0.06 | −4.30 | 258.2 | <0.001* |
| | SES | 0.15 | 0.04 | 4.11 | 253.8 | <0.001* |
| | Age | 0.06 | 0.04 | 1.65 | 204.7 | 0.101 |
| | Sex | 0.57 | 0.06 | 9.26 | 422.1 | <0.001* |
| | coTwin Sex | −0.11 | 0.06 | −1.69 | 422.1 | 0.092 |
| | **Between-Pair Effects** | | | | | |
| | Regular Bedtime | 0.02 | 0.04 | 0.58 | 89.4 | 0.566 |
| | Duration on School Nights | −0.02 | 0.04 | −0.43 | 158.7 | 0.666 |
| | Duration on Weekends | 0.01 | 0.04 | 0.27 | 190.8 | 0.785 |
| | Sleep Quality | 0.11 | 0.04 | 2.77 | 136.6 | 0.006* |
| | Daytime Sleepiness | 0.11 | 0.04 | 2.82 | 135.5 | 0.006* |
| | Snoring | 0.04 | 0.04 | 1.20 | 119.7 | 0.232 |
| | **Within-Pair Effects** | | | | | |
| | Regular Bedtime | 0.05 | 0.02 | 1.93 | 24.0 | 0.066 |
| | Duration on School Nights | −0.03 | 0.03 | −1.10 | 59.8 | 0.277 |
| | Duration on Weekends | 0.01 | 0.02 | 0.44 | 49.1 | 0.661 |
| | Sleep Quality | 0.08 | 0.03 | 3.03 | 95.6 | 0.003* |
| | Daytime Sleepiness | 0.06 | 0.03 | 2.01 | 66.1 | 0.048 |
| | Snoring | 0.02 | 0.03 | 0.66 | 75.7 | 0.513 |

* Indicates significant predictor after correcting for multiple testing, p value < .007.

**Table 6. Between and within models testing sleep measures as predictors of hyperactivity in grades 3, 5, 7, and 9.**

| Grade | Predictor | Std B | Std SE | t | df | p |
|---|---|---|---|---|---|---|
| **3** | Intercept | −0.19 | 0.06 | −3.36 | 301.0 | 0.001* |
| | SES | 0.13 | 0.04 | 3.53 | 261.6 | <0.001 |
| | Age | 0.11 | 0.04 | 2.90 | 222.0 | 0.004* |
| | Sex | 0.32 | 0.06 | 5.55 | 563.8 | <0.001* |
| | coTwin Sex | 0.05 | 0.06 | 0.86 | 563.6 | 0.388 |
| | **Between-Pair Effects** | | | | | |
| | Regular Bedtime | −0.01 | 0.04 | −0.30 | 81.4 | 0.765 |
| | Duration on School Nights | 0.01 | 0.06 | 0.25 | 194.9 | 0.806 |
| | Duration on Weekends | 0.03 | 0.05 | 0.56 | 215.4 | 0.575 |
| | Sleep Quality | 0.15 | 0.04 | 3.42 | 148.6 | 0.001* |
| | Daytime Sleepiness | 0.06 | 0.05 | 1.28 | 127.2 | 0.204 |
| | Snoring | −0.05 | 0.04 | −1.21 | 174.2 | 0.230 |
| | **Within-Pair Effects** | | | | | |
| | Regular Bedtime | 0.07 | 0.05 | 1.48 | 4.9 | 0.200 |
| | Duration on School Nights | 0.02 | 0.03 | 0.50 | 28.7 | 0.619 |
| | Duration on Weekends | 0.02 | 0.04 | 0.57 | 18.5 | 0.575 |
| | Sleep Quality | 0.04 | 0.02 | 2.42 | 81.5 | 0.018 |
| | Daytime Sleepiness | 0.03 | 0.02 | 1.37 | 56.7 | 0.175 |
| | Snoring | −0.02 | 0.01 | −1.63 | 116.3 | 0.106 |
| **5** | Intercept | −0.22 | 0.06 | −3.87 | 284.9 | <0.001* |
| | SES | 0.11 | 0.04 | 2.75 | 249.2 | 0.006* |
| | Age | 0.01 | 0.04 | 0.32 | 238.4 | 0.748 |
| | Sex | 0.39 | 0.05 | 7.26 | 578.4 | <0.001* |
| | coTwin Sex | 0.03 | 0.05 | 0.53 | 578.3 | 0.598 |
| | **Between-Pair Effects** | | | | | |
| | Regular Bedtime | 0.03 | 0.04 | 0.88 | 81.4 | 0.384 |
| | Duration on School Nights | −0.01 | 0.04 | −0.28 | 178.4 | 0.777 |
| | Duration on Weekends | 0.14 | 0.04 | 3.26 | 67.8 | 0.002* |
| | Sleep Quality | 0.20 | 0.04 | 4.54 | 107.1 | <0.001* |
| | Daytime Sleepiness | 0.02 | 0.04 | 0.42 | 126.8 | 0.677 |
| | Snoring | −0.03 | 0.04 | −0.78 | 133.3 | 0.439 |
| | **Within-Pair Effects** | | | | | |
| | Regular Bedtime | 0.05 | 0.04 | 1.46 | 9.5 | 0.176 |
| | Duration on School Nights | −0.02 | 0.02 | −0.87 | 74.1 | 0.387 |
| | Duration on Weekends | 0.03 | 0.02 | 1.25 | 34.5 | 0.221 |
| | Sleep Quality | 0.04 | 0.02 | 2.14 | 99.8 | 0.035 |
| | Daytime Sleepiness | −0.01 | 0.02 | −0.29 | 48.8 | 0.770 |
| | Snoring | 0.00 | 0.01 | 0.35 | 90.5 | 0.729 |

*(Continued)*

**Table 6.** (Continued)

| Grade | Predictor | Std B | Std SE | t | df | p |
|---|---|---|---|---|---|---|
| **7** | Intercept | −0.12 | 0.06 | −2.14 | 301.3 | 0.033 |
| | SES | 0.14 | 0.04 | 3.75 | 266.3 | <0.001* |
| | Age | 0.00 | 0.04 | −0.04 | 232.4 | 0.969 |
| | Sex | 0.31 | 0.05 | 5.63 | 608.1 | <0.001* |
| | coTwin Sex | −0.07 | 0.05 | −1.28 | 608.0 | 0.203 |
| | **Between-Pair Effects** | | | | | |
| | Regular Bedtime | 0.04 | 0.04 | 1.12 | 99.4 | 0.267 |
| | Duration on School Nights | −0.07 | 0.05 | −1.45 | 177.6 | 0.150 |
| | Duration on Weekends | 0.02 | 0.05 | 0.44 | 181.3 | 0.661 |
| | Sleep Quality | 0.11 | 0.04 | 2.39 | 181.8 | 0.018 |
| | Daytime Sleepiness | 0.09 | 0.04 | 1.94 | 131.0 | 0.054 |
| | Snoring | 0.01 | 0.04 | 0.14 | 140.1 | 0.888 |
| | **Within-Pair Effects** | | | | | |
| | Regular Bedtime | 0.02 | 0.02 | 0.94 | 15.2 | 0.362 |
| | Duration on School Nights | −0.04 | 0.02 | −1.69 | 47.8 | 0.098 |
| | Duration on Weekends | 0.05 | 0.02 | 2.28 | 56.9 | 0.027 |
| | Sleep Quality | 0.05 | 0.02 | 2.63 | 92.7 | 0.010 |
| | Daytime Sleepiness | 0.02 | 0.02 | 0.86 | 70.9 | 0.393 |
| | Snoring | 0.00 | 0.01 | 0.09 | 96.2 | 0.927 |
| **9** | Intercept | −0.23 | 0.06 | −3.80 | 258.3 | <0.001* |
| | SES | 0.13 | 0.04 | 3.19 | 253.5 | 0.002* |
| | Age | 0.04 | 0.04 | 1.01 | 203.8 | 0.315 |
| | Sex | 0.34 | 0.05 | 6.55 | 497.2 | <0.001* |
| | coTwin Sex | 0.10 | 0.05 | 1.83 | 497.2 | 0.068 |
| | **Between-Pair Effects** | | | | | |
| | Regular Bedtime | 0.04 | 0.04 | 1.03 | 89.4 | 0.307 |
| | Duration on School Nights | −0.04 | 0.05 | −0.86 | 158.7 | 0.389 |
| | Duration on Weekends | 0.03 | 0.04 | 0.73 | 190.8 | 0.468 |
| | Sleep Quality | 0.17 | 0.04 | 3.92 | 136.2 | <0.001* |
| | Daytime Sleepiness | 0.08 | 0.05 | 1.64 | 135.1 | 0.104 |
| | Snoring | 0.04 | 0.04 | 1.06 | 119.7 | 0.291 |
| | **Within-Pair Effects** | | | | | |
| | Regular Bedtime | 0.02 | 0.01 | 1.12 | 24.0 | 0.275 |
| | Duration on School Nights | −0.01 | 0.02 | −0.52 | 59.8 | 0.605 |
| | Duration on Weekends | 0.02 | 0.02 | 0.93 | 49.1 | 0.358 |
| | Sleep Quality | 0.04 | 0.01 | 2.85 | 95.5 | 0.005* |
| | Daytime Sleepiness | 0.02 | 0.02 | 1.32 | 66.1 | 0.192 |
| | Snoring | 0.02 | 0.01 | 1.24 | 75.7 | 0.218 |

* Indicates significant predictor after correcting for multiple testing, $p$ value <.007.

pairs who averaged more hyperactivity were associated with better mathematics achievement. This direction of effect was opposite to the zero order correlation between hyperactivity and mathematics, and was likely due to covariation among predictors in the regression model, in this case between inattention and hyperactivity. To test this explanation, an exploratory model was run including hyperactivity effects without the inattention effects. In this model, both the between- and within-pair effects for hyperactivity were significantly associated with mathematics in the expected direction, such that less hyperactivity predicted better mathematics.

An additional post-hoc analysis was conducted on the Grade 9 sample. In Grade 9, the within-pair effect for sleep quality was significantly associated with all four outcomes. Mediation analyses confirmed a significant indirect effect from within-pair differences in sleep quality to reading via within-pair differences in attention (indirect effect = 2.17, 95% CI [0.74, 3.94]), and no significant mediation occurred via within-pair differences in hyperactivity (indirect effect = −0.70, 95% CI [−2.32, .031]). = The total and direct effects for all predictors are in Supporting Information S2 File. After including the effects for inattention and hyperactivity, the within-pair effect for sleep quality was reduced in size and no longer significant. In summary, once inattention effects (both averaged for a pair and individual differences) were included, individual differences in daytime sleepiness no longer predicted reading or maths test performance in Grade 3 and individual difference in sleep quality no longer predicted reading or maths test performance in Grade 9.

## Discussion

One aim of this research was to add to the literature examining associations between children's sleep habits and sleep patterns and school performance in reading and mathematics. Previous research had identified reasonably modest relations between sleep and academic achievement overall, with a degree of inconsistency across the available results, from zero effects to moderately substantial ones. In the current study we used a community sample of students who had undertaken nation-wide academic tests from Grades 3–9, and parent-reported data on sleep habits and sleep patterns over this same age span. Additionally we aimed to examine the relations between sleep and ADHD since prior research has uncovered small but consistent relations between these two domains, with poorer sleep habits and patterns associated with poorer attention and more hyperactive behaviours. A post-hoc aim arose during the study: to identify whether inattention and hyperactivity mediated any sleep-school performance associations uncovered by the initial analyses.

Relations between sleep and reading and mathematics overall were weak. Of the sleep measures, only some of the raw correlations passed the multiple-testing significance criteria: Less snoring was associated with higher reading and mathematics performance in the two highest grades, 7 and 9; longer weekend sleep duration was associated with higher reading and mathematics in Grades 3, 5, and 7. When significant and in the expected direction of better sleep leading to better reading or mathematics, the correlations ranged from .07 to .12, indicating a very minor proportion of variance in reading and mathematics accounted for by sleep measures. One correlation, between sleep quality and reading in Grade 7, was in the unexpected direction of better quality sleep predicting poorer reading. This is an anomalous result when compared to the other grades and no association was evident in the between and within models. The only significant finding in the between and within-models was in Grade 9 when within-pair sleep quality was associated with better reading in the expected direction. Nonetheless, this effect disappeared when inattention was included. Thus we suspect that the zero order negative correlation between sleep quality and Grade 7 reading was biased by unmeasured confounders or influential cases that were accounted for in the full regression models but not in the correlations. Overall, the very modest relations identified here are in line with the general tenor of results reviewed in the Introduction. Variation in parent-reported sleep habits and sleep patterns, of the type and within the range captured in this study, are relatively independent of variation in reading and mathematics achievement as assessed by standardised tests administered in schools.

Results from the between- and within-pair models (Tables 3, 4), made possible through the use of twins, identified that better sleep quality predicted better mathematics and reading in Grade 9, and daytime sleepiness was related to poorer reading and mathematics amongst the youngest students, Grade 3. These observations emerged from within-pair

modelling, which has the advantage of controlling for family-level confounds. Overall these results mirror those reported by Musshafen et al. [6] in that relations between sleep and academic outcomes are small and inconsistent.

Relations between sleep habits and sleep patterns and inattention / hyperactivity were more consistent and stronger than those with academic performance, explaining up to 7% of the variance in hyperactivity in Grade 5, supporting findings from several comprehensive reviews [10–13]. These effects are considered medium [41] (e.g., a correlation of $r = .25$ is equivalent to explaining approximately 6% of variance calculated as partial eta-squared in the current study) and may potentially be more practically meaningful than the effect of sleep on academic performance. In other words, improvements in children's sleep habits and/or sleep patterns may be linked with fewer daytime hyperactive behaviours. A recent systematic review [42] indicates that effects of these sizes are similar to known risk factors including negative harsh discipline ($r = .19$), general maltreatment ($r = .30$) and parental incarceration ($r = .10$). Furthermore, and based on the raw correlations that reached significance (Table 2), sleep quality and daytime sleepiness were most consistently associated with inattention and hyperactivity in all grade levels. As was the case for reading and mathematics, snoring correlated with inattention and hyperactivity in the two highest grades, 7 and 9. The between and within models were partially consistent with the correlations (Tables 5, 6), although snoring was no longer associated with inattention and hyperactivity, and daytime sleepiness primarily related only to inattention, not hyperactivity.

Post-hoc mediation analyses were conducted in Grades 3 and 9. These were two situations where in the sensitive within models, one sleep variable related to both reading and/or mathematics and to inattention/hyperactivity: Grade 3 with daytime sleepiness and Grade 9 with sleep quality. These samples were smaller than the ones employed for computing correlations (Table 2) and the between and within models (Tables 3, 4) because of the reduced sample size for inattention and hyperactivity data (see Method). In the models with inattention and hyperactivity as covariates, the effects of individual differences in daytime sleepiness for Grade 3 and of sleep quality in Grade 9 on reading and mathematics were mediated by individual differences in inattention, and the direct effects were substantially reduced or essentially nullified (S2 File). In sum, inattention/ hyperactivity fully or partially accounted for observed relations between individual differences in sleep and reading or mathematics. In future research with, for example, other samples, other sleep assessment methods, and other academic domains, it would be prudent to routinely include measures of inattention/hyperactivity in view of the clear-cut picture evident in this and other research.

## Limitations

The sample used in this project did not deliberately include students with marked sleep difficulties, for example those receiving intervention in a clinical setting. Our results therefore do not speak to any effects such difficulties have on school performance. Relatedly, the majority of students in the sample sat at the "good" end of the possible scores in our scales—these Australian students on the whole are "good sleepers". This observation is interesting in its own right, given the sample size, but places a ceiling on the strength of relations between sleep parameters and academic performance through range restriction.

The fact that all student participants were members of multiple-birth sets, mostly twins but a few sets of triplets, may affect the reporting of sleep habits in that at least the same-sex sets may share a room more often than different-age siblings. However, our evidence indicates that most sleep patterns, and the ability of parents to rate members of twin pairs independently, did not generate within-pair correlations approaching unity—for sleep quality and snoring in each grade, correlations were around .55 and for daytime sleepiness, about .75. For bedtime regularity (capturing sleep habits), however, something plausibly more controlled by parents especially at younger ages, the correlation was .98 at Grade 3 and .88 at Grade 9. Similarly, bedtime regularity correlated .90 at Grade 3 and .85 at Grade 9. These high correlations impact the within-pair variance and the predictive power of this variable, but it does not necessarily reduce the between-pair variability. However, many families in out sample did endorse a high level of bedtime regulatity. We *a priori* decided on the variables to include in our model, in future, it might be well worth also deciding on a minimum amount of variation in a

predictor to include it in the models. It may be that bedtime regularity was a non-significant predictor in out analyses not because it is intrinsically unimportant, but because of ceiling effects.

One of the key limitations in this paper is the difficulty of accurately measuring sleep behaviours in children and adolescents using parent survey items. As noted above, this problem is not unique to the current study – indeed it impacts on much of the sleep research that relies on surveys of sleep [6]. Furthermore, while averaging responses to create the *sleep quality* and *daytime sleepiness* variables helps to attenuate multicollinearity between predictors, variability in all the sleep items was low overall, as pointed out earlier, leading to skewed and leptokurtic distributions. These issues affect the statistical power of the models. In addition, retrospective reports may be less accurate than concurrent reports of sleep habits; parents' reports of twins' sleep patterns may be less accurate than individuals reporting on themselves; and the accuracy of parent reports may decline as children age [43]. However, the overall agreement of the current results with much of the available literature increases confidence that the methods used are not creating unique problems of interpretation.

The stronger effects between sleep quality and inattention and hyperactivity, compared with those between sleep quality and reading and mathematics, might additionally be influenced by both predictor and outcome being reported by parents in the former models. This was an unavoidable feature of the project's design, using mail-outs to families country-wide and not to teachers, who could alternatively have provided inattention/hyperactivity assessments. Common method variance can inflate observed correlations between variables, though this can be offset by unreliability of the measurements [44]. Thus it is worth acknowledging that the findings of this study may be biased upwards (by common method variance) or downwards (by measurement unreliability).

Using standardised tests scores collected in schools avoids the same common method problem in the models using reading and mathematics as outcomes. However, the standardized tests that were used may not be close enough to day-to-day classroom performance to provide adequately nuanced information about students' achievement and thus these test scores may not be sensitive enough to be affected by sleep problems. Larger effects may be observed with more proximal assessments of classroom learning in comparison to standardised tests [45], or other, more reliable measures of sleep may show stronger associations with academic variables.

While not strictly a limitation, it needs to be noted that the samples, categorized by grade and academic domains, are not independent of each other. That is, they do not form independent replications of results at increasing grades because each of the grade samples comprises some of the same students. As a longitudinal study, there is some overlap in sample membership across grades, and of course the tests themselves, reading and mathematics, are each taken by each child.

## Conclusions

In these data, sleep habits and sleep patterns bore very modest relations with reading and mathematics performance in students across Grades 3–9, and the effects were not consistent across grades or tests. Sleep duration, particularly on weekends, related positively on some test occasions (longer duration relating to better school performance), but again not consistently. Bedtime regularity did not emerge as related to school performance at all. The results are in agreement with substantial portions of the recent literature, which also documents modest links for similar sleep and academic variables. In contrast, sleep measures correlated more strongly with inattention and hyperactivity. Sleep quality and daytime sleepiness were the standout predictors, emerging in most grades, particularly for inattention. In mediation models, the effects of sleep on the academic variables were fully mediated by inattention. It is plausible, therefore, that disturbed sleep influences school performance via issues in attention.

## Supporting information

**S1 File. Code and output for exploratory factor analyes.**
(PDF)

**S2 File. Supplementary tables.**
(DOCX)

**S3 File. Inclusivity in global research.**
(DOCX)

## Author contributions

**Conceptualization:** Katrina L. Grasby, Sally A. Larsen, Alice M. Gregory, Sarah Blunden, Juan J. Madrid-Valero, William L. Coventry, Brian Byrne, Richard K. Olson.

**Data curation:** Sally A. Larsen, William L. Coventry.

**Formal analysis:** Katrina L. Grasby.

**Funding acquisition:** William L. Coventry, Brian Byrne, Richard K. Olson.

**Investigation:** Katrina L. Grasby, Sally A. Larsen, Alice M. Gregory, Sarah Blunden, Katie S. Lewis, Brian Byrne, Richard K. Olson.

**Methodology:** Katrina L. Grasby, Sally A. Larsen, Juan J. Madrid-Valero.

**Project administration:** Sally A. Larsen, William L. Coventry.

**Resources:** William L. Coventry, Brian Byrne, Richard K. Olson.

**Software:** Katrina L. Grasby.

**Supervision:** Alice M. Gregory, William L. Coventry, Brian Byrne, Richard K. Olson.

**Validation:** Katrina L. Grasby, Juan J. Madrid-Valero.

**Writing – original draft:** Katrina L. Grasby, Sally A. Larsen, Brian Byrne.

**Writing – review & editing:** Katrina L. Grasby, Sally A. Larsen, Alice M. Gregory, Sarah Blunden, Katie S. Lewis, Juan J. Madrid-Valero, William L. Coventry, Brian Byrne, Richard K. Olson.

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
