## [Decision Letter · Decision Letter 0]

28 Jan 2026

PONE-D-25-42491Associations between Sleep Habits, Performance in Reading and Mathematics, and Inattention and HyperactivityPLOS One

Dear Dr. Larsen,

Thank you for submitting your manuscript to PLOS ONE. After careful consideration, we feel that it has merit but does not fully meet PLOS ONE’s publication criteria as it currently stands. Therefore, we invite you to submit a revised version of the manuscript that addresses the points raised during the review process.

We look forward to receiving your revised manuscript.

Kind regards,

Mohammad Hossein Ebrahimi

Academic Editor

PLOS One

Journal Requirements:

“Alice Gregory is an advisor for a project initially sponsored by Johnson’s Baby. She was a consultant for Perrigo (2021+). She receives royalties for two books Nodding Off (Bloomsbury Sigma, 2018) and The Sleepy Pebble (Flying Eye, 2019) and a sleep gift (The Gift of Sleep, Lawrence King Publishers, 2023). She was previously a CEO of Sleep Universal LTD (2022). She occasionally receives sample products related to sleep (e.g. blue light blocking glasses) and has given a paid talk to a business (Investec). She is a specialist subject editor at JCPP (sleep) for which she receives a small honorarium. She has contributed a paid article to Neurodiem.   ”

Please confirm that this does not alter your adherence to all PLOS ONE policies on sharing data and materials, by including the following statement: "This does not alter our adherence to  PLOS ONE policies on sharing data and materials.” (as detailed online in our guide for authors http://journals.plos.org/plosone/s/competing-interests If there are restrictions on sharing of data and/or materials, please state these. Please note that we cannot proceed with consideration of your article until this information has been declared.

“Australian Research Council Discovery Project DP120102414 and DP150102441”

5. Please note that funding information should not appear in any section or other areas of your manuscript. We will only publish funding information present in the Funding Statement section of the online submission form. Please remove any funding-related text from the manuscript.

Reviewer's Responses to Questions

**Comments to the Author**

1. Is the manuscript technically sound, and do the data support the conclusions?

Reviewer #1: Yes

Reviewer #2: Partly

2. Has the statistical analysis been performed appropriately and rigorously? 

Reviewer #1: I Don't Know

Reviewer #2: Yes

3. Have the authors made all data underlying the findings in their manuscript fully available?

The PLOS Data policy requires authors to make all data underlying the findings described in their manuscript fully available without restriction, with rare exception (please refer to the Data Availability Statement in the manuscript PDF file). The data should be provided as part of the manuscript or its supporting information, or deposited to a public repository. For example, in addition to summary statistics, the data points behind means, medians and variance measures should be available. If there are restrictions on publicly sharing data—e.g. participant privacy or use of data from a third party—those must be specified.requires authors to make all data underlying the findings described in their manuscript fully available without restriction, with rare exception (please refer to the Data Availability Statement in the manuscript PDF file). The data should be provided as part of the manuscript or its supporting information, or deposited to a public repository. For example, in addition to summary statistics, the data points behind means, medians and variance measures should be available. If there are restrictions on publicly sharing data—e.g. participant privacy or use of data from a third party—those must be specified.requires authors to make all data underlying the findings described in their manuscript fully available without restriction, with rare exception (please refer to the Data Availability Statement in the manuscript PDF file). The data should be provided as part of the manuscript or its supporting information, or deposited to a public repository. For example, in addition to summary statistics, the data points behind means, medians and variance measures should be available. If there are restrictions on publicly sharing data—e.g. participant privacy or use of data from a third party—those must be specified.requires authors to make all data underlying the findings described in their manuscript fully available without restriction, with rare exception (please refer to the Data Availability Statement in the manuscript PDF file). The data should be provided as part of the manuscript or its supporting information, or deposited to a public repository. For example, in addition to summary statistics, the data points behind means, medians and variance measures should be available. If there are restrictions on publicly sharing data—e.g. participant privacy or use of data from a third party—those must be specified.

Reviewer #1: Yes

Reviewer #2: Yes

4. Is the manuscript presented in an intelligible fashion and written in standard English?

Reviewer #1: Yes

Reviewer #2: Yes

5. Review Comments to the Author

Reviewer #1: The manuscript is methodologically sound and represents an important contribution, especially because of its large-scale twin design and simultaneous focus on sleep, ADHD, and academic outcomes. However, revisions are necessary to strengthen the interpretation, better address biases and enhance clarity. If these are addressed, the paper will provide a valuable and trustworthy addition to the field.

Reviewer #2: This manuscript examines associations between several aspects of sleep habits and academic performance (reading and mathematics), as well as inattention and hyperactivity, in a large sample of Australian twins. The use of nationwide standardized academic assessments and a twin-based between–within modeling approach are notable strengths. Overall, the main conclusions—namely that associations between sleep habits and academic outcomes are generally small, while relations with inattention/hyperactivity are more consistent—are supported by the data.

Comments and suggestions for improvement

1. Sleep measurement and retrospective reporting

A proportion of the sleep data was collected retrospectively. While this is understandable in a longitudinal design, it would be helpful if the authors could briefly clarify how retrospective sleep reporting was implemented and acknowledge more explicitly how this may have influenced the results. If feasible, a short sensitivity analysis or descriptive comparison between retrospective and concurrent reports would further strengthen the paper.

2. Parent-reported measures

Both sleep habits and inattention/hyperactivity were reported by parents, which may contribute to somewhat stronger associations between these domains compared with academic outcomes. This point is already noted in the limitations section, but a slightly expanded discussion would be helpful for readers.

3. Sleep habit composites

The selection of specific items from the Children’s Sleep Habits Questionnaire and the creation of composite measures for sleep quality and daytime sleepiness are reasonable. Providing a small amount of additional detail on the factor-analytic justification for these composites (for example, in supplementary material) would improve transparency and reproducibility.

4. Within-pair effects

The between–within modeling approach is a major strength of the study. For some sleep variables that show very high similarity within twin pairs (e.g., bedtime regularity), it may be useful to note that limited within-pair variability could reduce the precision of within-pair estimates, and therefore, such effects should be interpreted cautiously.

5. Unexpected or inconsistent findings

A small number of associations were in an unexpected direction or varied across grades. The authors appropriately note the overall inconsistency of effects; a brief additional comment suggesting possible explanations (e.g., measurement issues or statistical suppression) would further aid interpretation.

6. Data availability

The authors clearly state that reading and mathematics data are owned by Australian state and territory governments and cannot be shared publicly. It may be helpful to briefly indicate whether and how qualified researchers can request access to these data through the relevant authorities.

Summary

In summary, this is a well-conducted study with a strong design and a large, informative dataset. Addressing the points above would further clarify the methodology and interpretation, but the overall findings are sound and consistent with much of the existing literature.

6. PLOS authors have the option to publish the peer review history of their article (what does this mean?). If published, this will include your full peer review and any attached files.). If published, this will include your full peer review and any attached files.). If published, this will include your full peer review and any attached files.). If published, this will include your full peer review and any attached files.

...

Reviewer #1: **Yes:** Fathi Ali ArayeFathi Ali ArayeFathi Ali ArayeFathi Ali Araye

Reviewer #2: **Yes:** Mohammad EslamiMohammad EslamiMohammad EslamiMohammad Eslami

You may also use PLOS’s free figure tool, NAAS, to help you prepare publication quality figures: https://journals.plos.org/plosone/s/figures#loc-tools-for-figure-preparation

---

## [Author Response · Author response to Decision Letter 1]

25 Mar 2026

Please see our response to reviewers' comments in the attached document.

---

## [Editor Report · Decision Letter 1]

8 Apr 2026

Associations between Sleep Habits, Performance in Reading and Mathematics, and Inattention and Hyperactivity

PONE-D-25-42491R1

Dear Dr. Larsen,

We’re pleased to inform you that your manuscript has been judged scientifically suitable for publication and will be formally accepted for publication once it meets all outstanding technical requirements.

Kind regards,

Mohammad Hossein Ebrahimi

Academic Editor

PLOS One
---

## [Editor Report · Acceptance letter]

PONE-D-25-42491R1

PLOS One

Dear Dr. Larsen,

I'm pleased to inform you that your manuscript has been deemed suitable for publication in PLOS One. Congratulations! Your manuscript is now being handed over to our production team.

Kind regards,

on behalf of

Dr. Mohammad Hossein Ebrahimi

Academic Editor

PLOS One